# A Novel Elastic Sensor Sheet for Pressure Injury Monitoring: Design, Integration, and Performance Analysis

**Mohammad Mohammad Amini** [1,*], **Mohammad Ghassem Farajzadeh Devin** [1], **Paulo Alves** [2,3], **Davood Fanaei Sheikholeslami** [1], **Fatemeh Hariri** [1], **Rogério Dionísio** [4,5], **Mahdi Faghihi** [1], **Fernando Reinaldo** [4,5], **José Carlos Metrôlho** [4,5] **and Luis Fonseca** [4]

[1] Sensomatt Lda., R&D Department, 6000-767 Castelo Branco, Portugal;
m.farajzadeh@sensomatt.com (M.G.F.D.); d.fanaei@sensomatt.com (D.F.S.); f.hariri@sensomatt.com (F.H.);
m.faghihi@sensomatt.com (M.F.)

[2] School of Nursing, Universidade Católica Portuguesa, Rua de Diogo Botelho, 1327, 4169-005 Porto, Portugal;
pjalves@ucp.pt

[3] Wounds Research Lab—Centre for Interdisciplinary Research in Health, Institute of Health Sciences,
4200-072 Porto, Portugal

[4] School of Technology, Polytechnic Institute of Castelo Branco, 6000-081 Castelo Branco, Portugal;
rdionisio@ipcb.pt (R.D.); fribeiro@ipcb.pt (F.R.); metrolho@ipcb.pt (J.C.M.); f.luis@ipcbcampus.pt (L.F.)

[5] DiSAC—Research Unit on Digital Services, Applications and Content, 6000-767 Castelo Branco, Portugal

\* Correspondence: mm_amini@sensomatt.com; Tel.: +351-913676941

**Abstract:** This study presents the SENSOMATT sensor sheet, a novel, non-invasive pressure monitoring technology intended for placement beneath a mattress. The development and design process of the sheet, which includes a novel sensor arrangement, material selection, and incorporation of an elastic rubber sheet, is investigated in depth. Highlighted features include the ability to adjust to varied mattress sizes and the incorporation of AI technology for pressure mapping. A comparison with conventional piezoelectric contact sensor sheets demonstrates the better accuracy of the SENSOMATT sensor for monitoring pressures beneath a mattress. The report highlights the sensor network's cost-effectiveness, durability, and enhanced data measurement, alongside the problems experienced in its design. Evaluations of performance under diverse settings contribute to a full understanding of its potential pressure injury prediction and patient care applications. Proposed future paths for the SENSOMATT sensor sheet include clinical validation, more cost and performance improvement, wireless connection possibilities, and improved long-term monitoring data analysis. The study concludes that the SENSOMATT sensor sheet has the potential to transform pressure injury prevention techniques in healthcare.

**Keywords:** pressure injuries; sensor sheet; elastic sensors; pressure measurement; electronic design

## 1. Introduction

Pressure injuries represent a significant healthcare challenge due to their high prevalence, particularly among individuals with limited mobility such as the elderly or those with physical impairments. They not only are painful and potentially life-threatening but also contribute to extended hospital stays and increased healthcare costs [1]. Among the many methods explored, the use of sensor technology in combination with AI-based decision making has emerged as a promising approach for preventing pressure injuries. This technology aims to monitor pressure points continuously, thereby enabling early detection of potential injury formation and prompt intervention [2]. However, many of the currently available sensor systems are limited by factors such as cost, comfort, accuracy, and ease of use, which restrict their widespread adoption in routine clinical practice.

Pressure injury prevention remains a challenge within the healthcare community despite the study of various preventive measures. A common theme across many studies

is the importance of frequent changes in a patient's position to reduce prolonged pressure on a specific area of the body, known as repositioning or turning. However, the application of this approach can be labor-intensive, placing a significant burden on healthcare workers and caregivers.

There is a growing interest in the investigation of sensors put into or beneath mattresses for applications such as bed fall detection, sleep posture monitoring, chest activity tracking, seizure detection, and sleep disorder diagnosis. According to [3–8], some studies focused on detecting vibration levels to detect heart failure symptoms, while other studies included microphones to diagnose illnesses such as snoring and sleep apnea. However, none investigated the possibility of a novel pressure mapping system employing a specialized sensor sheet to detect maximum pressure and predict the danger of pressure damage.

Another scientific method [9–11] focused on flexible, wearable (or attachable) sensors for direct assessments of bodily touch. There are temperature and vibration sensors for monitoring sleep, pressure sensors in gloves for human–machine interfaces, and neuroelectric activity sensors for physiological monitoring. Although these sensor systems are promising, their intrusive nature characterizes them as higher-risk medical devices, restricting their applicability to large patient groups and general consumers. In contrast, the SENSOMATT sensor sheet does not touch users' bodies or interfere with medical operations, making it a less invasive alternative.

In the framework of pressure injury prevention, solutions from many domains are being pursued. The measurement of body-induced pressure is a separate field of study from mattress sensor research. Pressure injuries, which are frequently caused by sustained pressure on a particular body region, offer a substantial health concern, particularly for those with limited mobility. Developing non-invasive, accurate sensors to detect early indicators of pressure injuries could change preventative healthcare and dramatically enhance patients' quality of life.

S. Tsuchiya et al. (2016) investigated the use of small changes (SCs) in body position to alleviate pressure on vulnerable areas, notably the sacral region [12]. These SCs mimic the natural shifts in body position that occur spontaneously during sleep, thus providing a non-disruptive, feasible approach for pressure redistribution. Tsuchiya and colleagues identified several SC combinations that resulted in optimal pressure redistribution, particularly when the buttock region was involved, and suggested the use of a specially designed air mattress to automate the process. However, their study was limited by the focus on healthy young women, suggesting the need for further research involving other demographics with higher pressure injury risk, such as the elderly or patients with physical impairments.

The specific role of technology in facilitating pressure injury prevention has also been explored. For instance, Yu-Wei Liu et al. (2015) developed a bed-centered telehealth system (BCTS) using a motion-sensing mattress, WhizPAD, which captures valuable data about the bed occupant's physical activities [13]. The bed-based activity detection included monitoring of on/off bed instances, sleep posture, movement counts, and respiration rate, all of which could provide valuable insights into the risk of pressure injury development. However, the cost and complexity of such systems, as well as concerns about comfort, may limit their widespread adoption in routine clinical practice. In our work on the SENSOMATT sensor sheet, we were particularly concerned with developing a cost-effective solution that may be used beyond the limitations of costly intensive care unit (ICU) beds. Our objective was to make this technology available for use in nursing homes and by patients getting care at home who are at risk for incurring pressure injuries. This focus on cost distinguished us from other market alternatives and had a substantial impact on our electronic development plans.

In a more targeted approach, D. Pickham et al. (2018) explored the use of wearable patient sensors in preventing hospital-acquired pressure injuries in critically ill adults in ICUs [14]. Their pragmatic randomized clinical trial demonstrated a significant reduction in pressure injury occurrence and increased turning compliance in the group that utilized wearable patient sensors. Despite their promising findings, they acknowledged several

study limitations, including incomplete data on support surfaces and potential external validity threats due to treatment diffusion. Concerns exist about potential skin damage due to prolonged sensor contact, questioning the long-term safety and effectiveness of the sensor. The primary challenge is ensuring that the sensor does not inadvertently cause skin injuries or irritations while preventing pressure injuries. While our foremost priority is pressure injury prevention, the preservation of skin health and user comfort is equally vital. Consequently, the SENSOMATT sensor sheet's design, materials, and functionality require continuous refinement based on comprehensive studies to optimize user benefit and minimize potential risks.

The field of pressure sensor technology has seen remarkable advancements over the years, aiming to address the persistent healthcare challenge of pressure injuries. Recent works in the domain has centered around the development of sensor sheets that allow for constant pressure monitoring, providing potential avenues for early intervention in pressure injury cases. Notable amongst these advancements is the work of C. Yan et al. (2012), who focused on the development of tactile sensors based on fiber Bragg grating (FBG) sensing elements [15]. The researchers were able to construct a sensor array embedded within a polymer sheet capable of detecting and characterizing pressure changes. These sensors utilize optical wavelength-encoded sensing signals, which show promise in applications such as pressure mapping and tactile sensing. Additionally, these FBG sensors demonstrated immunity to electromagnetic interference, absence of local electric current, and environmental ruggedness, all crucial factors in healthcare settings. Yan et al.'s flexible pressure sensing sheets indicate the potential benefits of integrating these sensors into materials such as mattresses for constant pressure monitoring. However, pressure- and temperature-sensing techniques are primarily designed for contact measurement, which is beyond the scope of this study. Contact pressure sensing can increase the friction and temperature of the skin, which may result in pressure injuries. Consequently, such methods are rarely implemented in practice.

Concurrent advancements in pressure sensor technology have explored the utilization of textiles as a base material. C. Gumus et al. (2022) presented a novel, scalable technique for manufacturing textile-based pressure sensor arrays [16]. These sensors, characterized by their softness, flexibility, and breathability, offer potential applications such as human motion recognition and object detection. The design involved conductive knit fabric electrodes and thermoplastic polyurethane (TPU) sheets serving as dielectric layers. The researchers were able to create scalable air gaps between the layers, allowing the capture of low pressures of less than 1 kPa. Additionally, the use of multi-layer TPU sheets extended the working range of the sensors to 1000 kPa, showcasing the potential for these sensors to be used in high-pressure applications. The work of Gumus et al. suggests the possibility of incorporating textile-based sensors into healthcare applications, such as pressure injury prevention, where their softness and breathability can enhance patient comfort. Even considering factors such as softness and patient comfort, touch sensors are seldom used in ICU departments for pressure measurement and risk analysis of pressure injuries, unless they are explicitly deployed for academic research.

Lastly, E. B. Monroy et al. (2020) proposed an intelligent, real-time monitoring system using wearable inertial sensors attached to the patients' clothing [17]. This system captured data on body posture and orientation, which were then processed using a data-driven model to classify postures. It also incorporated a knowledge-based fuzzy model to calculate the priority of postural changes based on expert-defined protocols. Despite several implementation challenges, this system demonstrated the potential for real-time, non-invasive monitoring of postural changes to prevent pressure injuries.

The literature suggests a growing interest in the application of sensor technology in pressure injury prevention. However, the cost, comfort, and ease of use of such technologies remain key challenges that need to be addressed. Therefore, our study, focusing on the design and integration of a cost-effective, non-invasive, and user-friendly sensor sheet, SENSOMATT, represents a significant contribution to this field of research.

The present study introduces a novel sensor technology, the SENSOMATT sensor sheet, specifically designed to address these limitations. The SENSOMATT sensor sheet incorporates an array of pressure sensors embedded in an elastic rubber sheet that can be placed underneath a mattress. This unique configuration allows for continuous pressure measurement in a non-invasive manner, offering potential benefits in terms of comfort and convenience. Importantly, the sensor sheet can be calibrated easily when a different mattress is used. The key contributions of this paper are the detailed design and integration of the SENSOMATT sensor sheet, including the materials used, sensor arrangement, and network design.

Due to the considerable displacement of the elastic cap on the SENSOMATT sensor sheet, it is possible to utilize it in contact with flexible fabrics and accurately record even minute pressures. In contrast to conventional piezoelectric contact sensor sheets, which excel in direct contact applications but lack accuracy when sensing pressures from beneath a mattress, the SENSOMATT sensor sheet excels in both areas. To show the efficacy of SENSOMATT, we compared pressure measurements between our sensor sheet and a commercial piezoelectric sensor sheet. Additionally, the SENSOMATT sensor sheet is cost-effective, with a total price that is less than one-fifth that of competitive pressure sensor sheets. It incorporates AI technology for pressure mapping and measures individual pressure locations using 72 cells. In conjunction with our AI system, we present an overview of pressure distribution and pinpoint crucial pressure sites for forecasting the risk of developing a pressure ulcer. The AI model was taught using data gathered from 60 volunteers sleeping in 28 frequent ICU patient postures.

In the hardware selection phase for the SENSOMATT sensor sheet, we focused on cost-effectiveness and excellent functionality. We present a complete report on the performance of the sensor sheet network and conduct a comparative analysis of pertinent designs, emphasizing hardware constraints and providing a thorough discussion on how to circumvent them.

The remaining sections of this study are organized methodically in order to provide a comprehensive overview of our findings. In Section 2, we delve into the design of the sensor cell and sheet, shining light on the fundamental reasons that influenced our design decisions, as well as the technological elements that enhance the operation of the sensor cells and sheets.

The third section is devoted to the design of the sensor network, discussing the obstacles encountered and the solutions applied. We address the connectivity of the sensor network, the employed communication protocols, and their effect on the system's performance and dependability. In addition, we describe the obstacles encountered during the design and testing phases, including hardware limitations and software compatibility difficulties, and how we overcame them to improve overall system performance.

In Section 4, we provide our findings and conduct a comprehensive analysis. This includes a thorough evaluation of the effectiveness of our design decisions, the functionality of the sensor network, and any potential real-world ramifications. In addition, we analyze the functioning of the SENSOMATT sensor sheet under different environmental circumstances and measure the device's adaptability and responsiveness.

This research aims to provide a comprehensive understanding of the SENSOMATT sensor sheet, its design process, prospective applications, and future development directions.

## 2. The Proposed Sensor Cell and Sheet Design

The development of the SENSOMATT sensor sheet involved a multi-step process, including the selection of materials, sensor design and testing, sensor arrangement, and integration of the sensors and network into the elastic rubber sheet.

Material selection: The material used for the sensor sheet needed to be both flexible and durable to withstand regular use. After testing several materials, synthetic rubber was chosen due to its excellent elasticity, resistance to wear and tear, and cost-effectiveness. Moreover, it was found that a thickness of 1 mm for the sensor caps provided the optimal

balance between flexibility and durability. The different tested cap materials can be seen in
Figure 1.

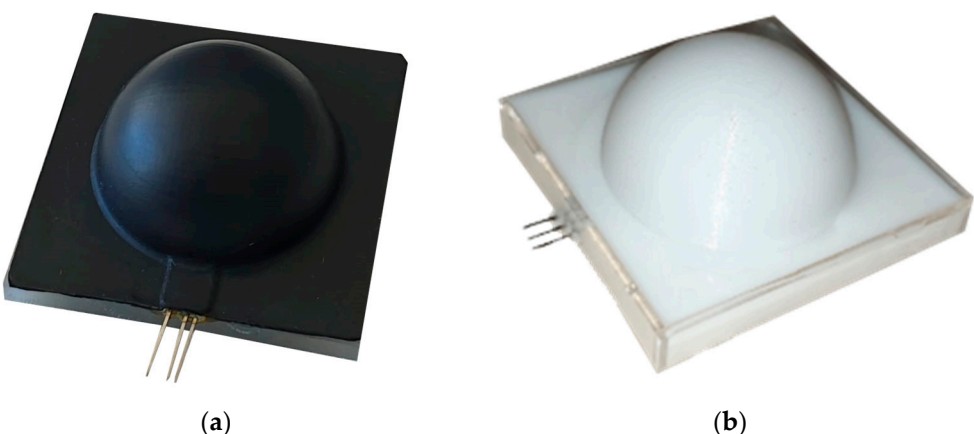

(**a**)                                                                                    (**b**)

**Figure 1.** Different materials used for sensor cell cap production: (**a**) synthetic vinyl-based rubber;
(**b**) silicon rubber.

Sensor selection and testing: The SENSOMATT sensor sheet contains a matrix of
pressure sensors that detect changes in pressure when a person lies or sits on the mat-
tress. Various types of pressure sensors were tested, including piezoelectric, piezoresistive,
and capacitive sensors. The best performance was obtained with piezoresistive sensors,
which exhibited superior sensitivity and accuracy. The analog absolute pressure sensor
KP236N6165 was ultimately chosen, and the sensor cell module was developed appro-
priately. The KP236N6165 is equipped with high-precision pressure sensing (1.0 kPa), a
ratiometric analog output, a wide temperature range ($-40\,^\circ$C to 125 $^\circ$C), and a calibrated
transfer function that translates the pressure range of 60 kPa to 165 kPa into a voltage range
of 0.2 V to 4.8 V.

On the sensor sheet, the sensors were ultimately organized in a 12 by 6 array. This
design provides extensive coverage of pressure points, ensuring accurate and consistent
pressure measurement. Sensomatt Lda. (2021) [18] proposed the concept of deploying
flexible pressure sensors in the shape of bubbles that operate separately and are networked
under the mattress. This idea is depicted in Figure 2. In this invention, a wireless network
was sought, but financial constraints pushed us to employ a cable connection for the
current project.

Integration of sensors and network: The sensors were connected to an all-element
electronic board, which transmitted the pressure data to a master board. The master board
was responsible for processing the data and communicating with the outside world. The
design of this network was a crucial aspect of the SENSOMATT sensor sheet, as it had to
ensure quick and reliable data transfer. A proprietary network was developed that could
connect over 100 sensor cells in just a few seconds.

Throughout the development process, the SENSOMATT sensor sheet underwent
rigorous testing to evaluate its durability, accuracy, and sustainability. These tests confirmed
that the sensor sheet was not only robust and accurate but also environmentally friendly,
contributing to its potential suitability for widespread use in healthcare settings.

### 2.1. Sensor Cell Electronic Design

The final design of the sensor cell effectively harnesses the capabilities of the AT-tiny
microcontroller, thereby achieving an optimal balance between power consumption and
processing speed. The AT-tiny13A, a low-cost, low-power CMOS 8-bit microcontroller
based on the AVR enhanced RISC architecture, executes powerful instructions in a single
clock cycle. This functionality allows for impressive throughputs nearing 1 MIPS per MHz,
enabling the sensor cell to perform high-endurance tasks with minimal power consumption.

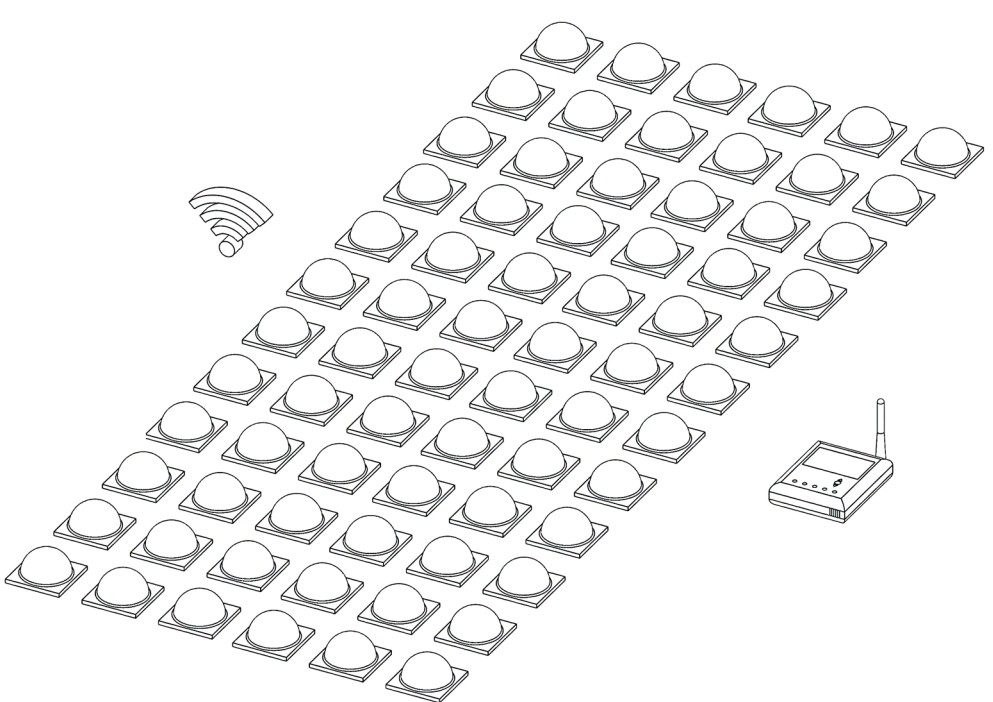

**Figure 2.** Patented array of sensors to measure the force and pressure from a place under the mattress. Each cell works separately in a harmonic network of sensors.

The sensor cell PCB design and the assembled electronic board can be observed in Figure 3. Figure 3a represents the circuit design created using Altium software, Figure 3b showcases the two-layer green laminated board, and Figure 3c depicts the sensor cell enclosed within the sensor sheet.

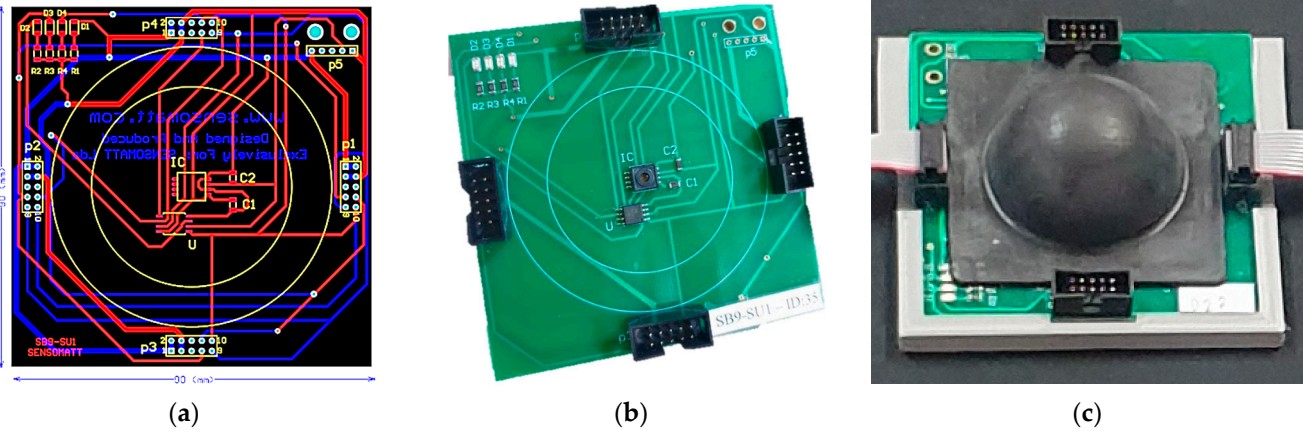

(**a**)　　　　　　　　　　　　(**b**)　　　　　　　　　　　　(**c**)

**Figure 3.** (**a**) The sensor cell designed PCB, (**b**) the printed two-layer PCB with green laminate cover, and (**c**) the final sensor cell with rubber cap.

The sensor cell final design, as demonstrated through circuit and PCB layouts, underscores its versatility and potential for scaling, marking a promising advancement in the field of pressure sensor technology.

### 2.2. Sensor Sheet Electronic Design

In the development of SENSOMATT, a critical facet of design and innovation was the sensor sheet. Its structure, arrangement, and integration played a pivotal role in making possible the unique feature of the product—the ability to measure pressure from underneath a mattress, only requiring recalibration for different mattresses. Both the mechanical and

the electrical aspects of the sensor sheet design are integral to the overall operation of the system and, hence, were addressed with significant attention and precision.

The sheet, primarily composed of synthetic rubber, underwent substantial testing for determining the ideal thickness. A thickness of 0.5 mm was finally settled upon, striking a balance between the sturdiness of the sheet and its ability to flex and conform to the mattress's shape. Plastic plugs were employed to secure the sensors to the sheet, ensuring the stability and reliability of sensor readings.

The sensor sheet constitutes a 72-sensor electrical board arranged in a 12 × 6 array. The sensor selection and arrangement were approached with meticulous precision, tested through several iterations of arrangements for durability, accuracy, and sustainability. The arrangement of 40 sensors within a 5 mm sheet was found to exhibit the best performance. The sensor integration into the sheet also demanded a thorough understanding of the material and its characteristics.

The sensor signals are acquired through a master board specifically designed to be compatible with Arduino Mega, utilizing its I/O ports, internal buffer, and processor capabilities. Figure 4 shows the circuit design and the PCB of the master board.

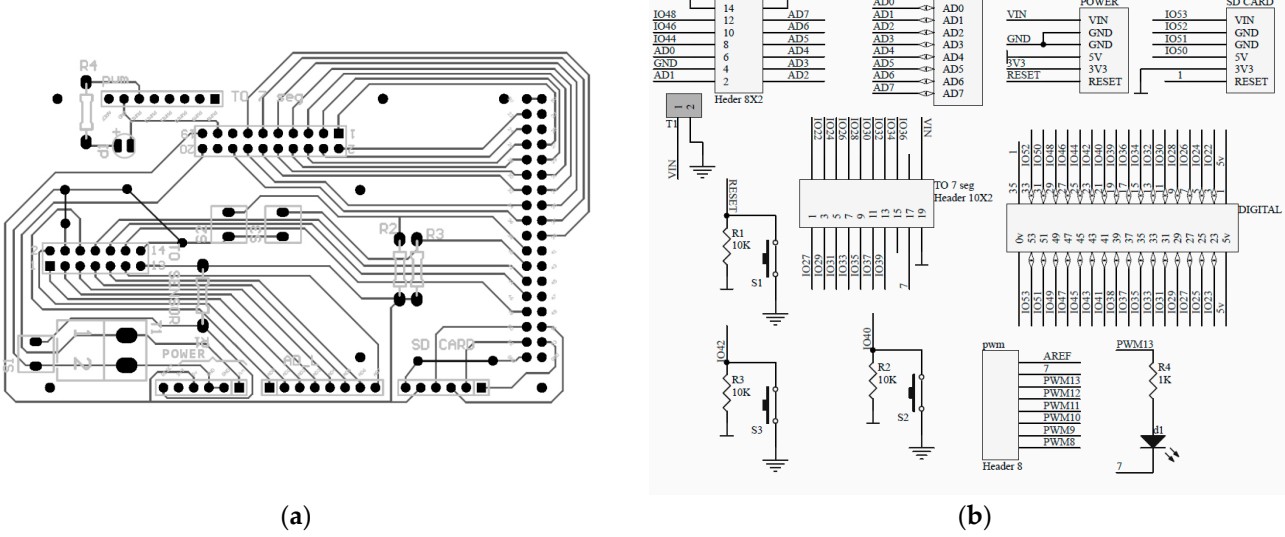

(**a**)     (**b**)

**Figure 4.** Circuit design of the master board: (**a**) the PCB; (**b**) the schematic.

The development of the sensor sheet was a nuanced and iterative process, integrating elements of electrical design, material science, and network architecture. The sheet's final design represents the culmination of rigorous testing and innovative thinking, laying a solid foundation for the unique capabilities of SENSOMATT.

## 3. The Proposed Network Design and Integration

A significant contribution to the overall functioning of SENSOMATT is the design and integration of the network within the sensor sheet. The network design aims to facilitate seamless communication between the sensors and the master board, thus ensuring accurate and timely data transfer.

The integration of the sensor network within the sheet marked a significant phase in the design process. The network was created to be adaptable to any configuration or dimension. This cost-effective proprietary network is capable of rapidly connecting over 100 sensor cells, enabling the collection and transmission of real-time data.

The restrictions of the AT-tiny13A IC represented one factor that influenced the network protocol decision. It has few communication ports, forcing the development of a novel network communication protocol despite being powerful and cost-effective. Consequently, a unique network protocol was devised to be compatible with the AT-tiny13A microcontroller-based final sensor cell module.

The AT-tiny13A microcontroller within the sensor cell connects with the master board using the custom-designed serial network. Due to the sensor cell's four parallel ports that can link in any direction, the network can be constructed in a variety of ways, as the network protocol does not restrict the shape of connections.

In this network, more than 100 sensor cells can communicate with the master board in less than one second. The network connections comprise six signals: VCC, GND, CLK, RST, RPT, and DAT.

GND and VCC: The intended supply voltage is 5 volts. Note that the proposed network structure can operate with multiple voltage levels, assuming compatible hardware.

CLK, RST, and RPT: The network master controls these three signals. Each slave (sensor cell) on the network is assigned a unique identifier. When the slave ID matches the CLK count, transmission of the slave's ID and sensor value to the master begins. The RST signal resets the CLK count. If the master transmits an RPT signal, the chosen slave will repeat its data on the network.

This signal is created by network slaves in order to transmit data to the master. Notably, all slaves' DAT signals are configured as pull-up digital inputs. When a slave wishes to transmit its data, it switches its configuration to a digital output and then back to a pull-up digital input.

Notably, because the AT-tiny13A microcontroller lacks a crystal clock, it cannot communicate flawlessly via a serial network. Small oscillator timing fluctuations can result in data loss; hence, each slave returns two data points every round. The first is constant data (low-level voltage) with a 300-microsecond length, and the second is data (second low-level voltage) whose duration is dependent on the output value of the sensor cell. Consequently, the master measures the pulse width using accurate internal clocks. As previously explained, for each slave, two pulses are created; the first pulse is used for calibration, and the second is utilized to record the selected sensor cell pressure measurement. Figure 5 depicts the chronology of the proposed network protocol.

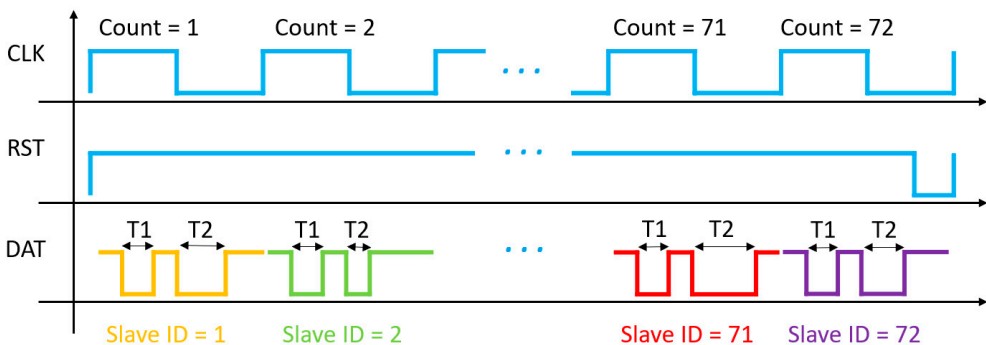

T1: Time of Calibration pulse (whose width is already known and is equal to 300 $ms$)

T2: Time of data pulse(whose width is related to the pressure value)

**Figure 5.** The timeline of the proposed network protocol.

The network was carefully integrated into the sensor sheet to ensure that it aligned with the sheet's mechanical and physical properties. By taking into consideration the sheet's flexibility and the positioning of the sensors, the network was seamlessly embedded within the sheet to ensure optimal functionality.

Through careful design considerations and a rigorous testing process, the network integration within the SENSOMATT sensor sheet enhances the system's performance and provides a reliable method for gathering and processing pressure data. The comprehensive design and integration process led to the creation of a network that not only meets the system's requirements but also bolsters its overall functionality.

## 4. Results and Discussion

### 4.1. Practical Results

The SENSOMATT sensor sheet is designed for usage in hospitals, clinics, and nursing homes, making its interoperability with these settings essential. Field experiments and data gathering undertaken in actual working conditions demonstrate the adaptability of the SENSOMATT sensor sheet, demonstrating its capacity to operate well in a variety of settings. There are several supporting publications showing that, in general, data collected from the sensor sheet provided valuable insights into pressure distribution patterns and allowed for the identification of high-pressure areas that could lead to the development of pressure injuries [19].

In order to assess the repeatability of the data and calibrate the sensor sheet, scales ranging from 1 to 5 kg were positioned in 12 distinct locations atop the mattress. Each position was examined 10 times, resulting in 72 sensor cell data values per examination. As a result, 600 tests were run, from which 43,200 data points were gathered. Figures 6 and 7 depict a few of these test outcomes. In particular, Figure 6 depicts repeated data taken from each sensor throughout numerous tests; Figure 6a corresponds to the 1 kg load in position 6, and Figure 6 corresponds to the 5 kg load in position 10. Figure 7 depicts the mean results from several testing for 12 unique sites on the mattress, with Figure 7a exhibiting the top view of the 1 kg weight, and Figure 7b depicting the 3D image of the 5 kg load. These typical graphs demonstrate the dependability and consistency of the sensor sheet.

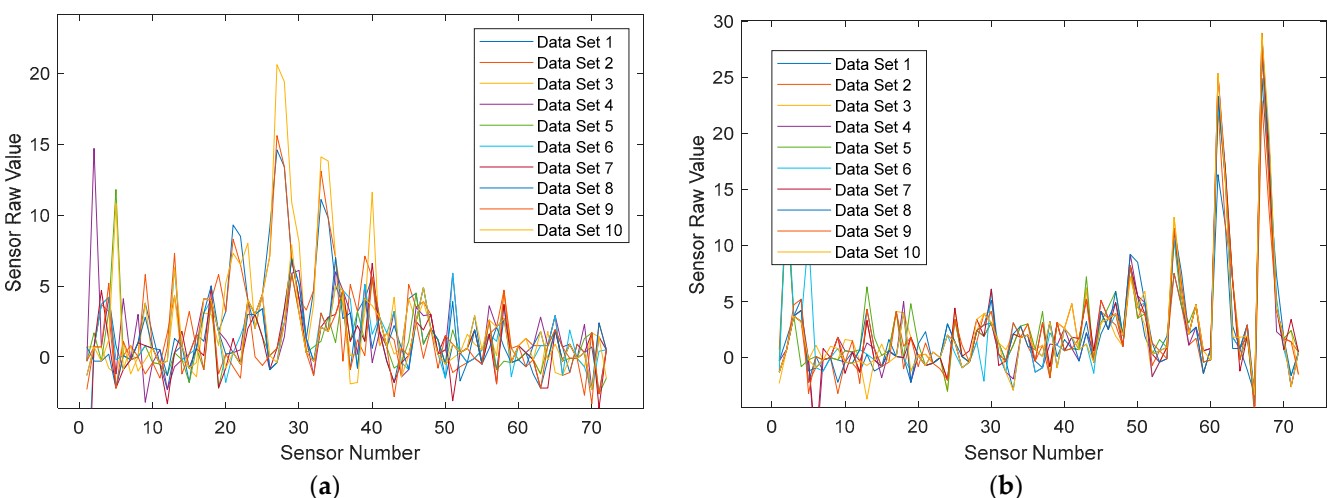

**Figure 6.** Measurements obtained from each sensor during multiple tests: (**a**) the 1 kg load in position 6; (**b**) the 5 kg load in position 10.

### 4.2. Durability and Reliability

The durability and reliability of the sensor sheet were assessed to ensure its long-term functionality in real-world scenarios. In this work, finite element analysis was employed for design help. In a mathematical model utilizing commercial finite element methods, multiple sensor cell shapes were created and subjected to varied load circumstances. The approach and results are published separately. Figure 8 provides an example of the data, displaying the mesh study of the sensor cells and enabling a comparison of stress fluctuation across different cells under varying load conditions. The final form of the sensor cells was decided on the basis of these models and subsequent mechanical tests.

A test was developed for each sensor cell in order to examine the repeatability of the sensor output voltage, checking the output voltage with scales ranging from 100 gr to 500 gr. Signals were subjected to appropriate low-pass filters according to the results. The reactions to varied loads are illustrated in Figure 9 as a depiction of these outcomes.

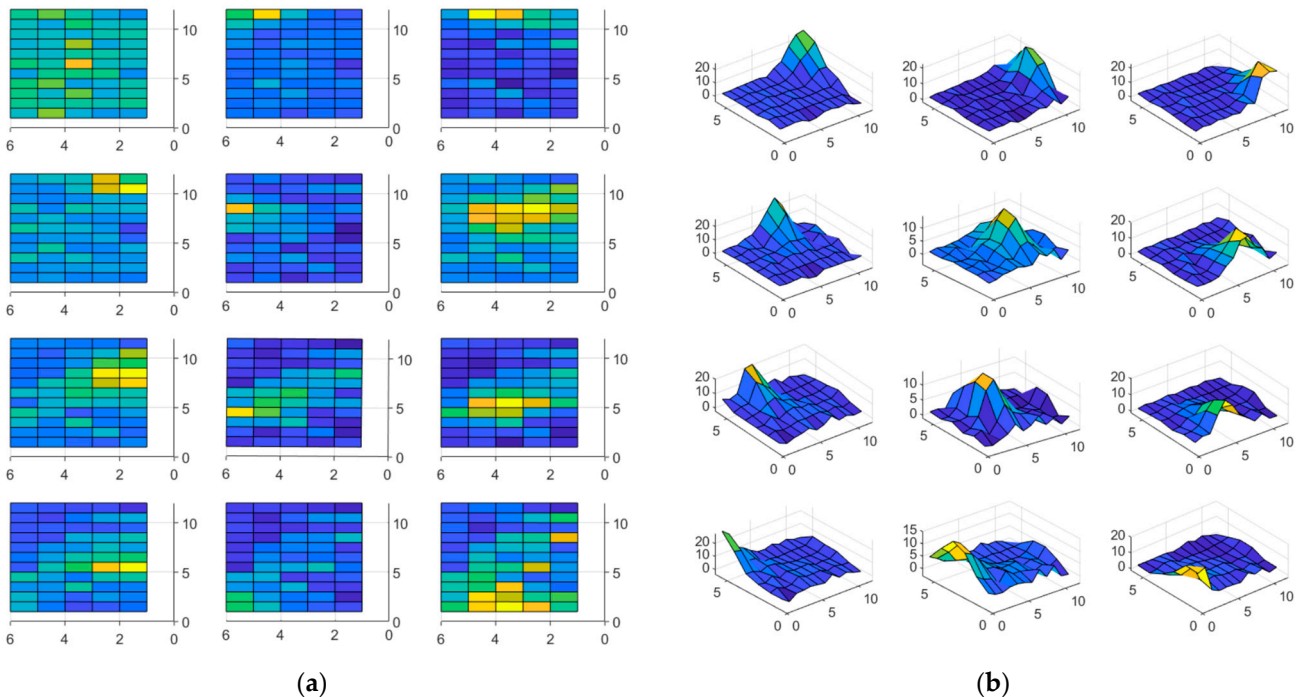

(a)          (b)

**Figure 7.** The value of several tests for 12 different points (symmetrical positions) on the mattress: (**a**) 1 kg load top view; (**b**) 5 kg load 3D view.

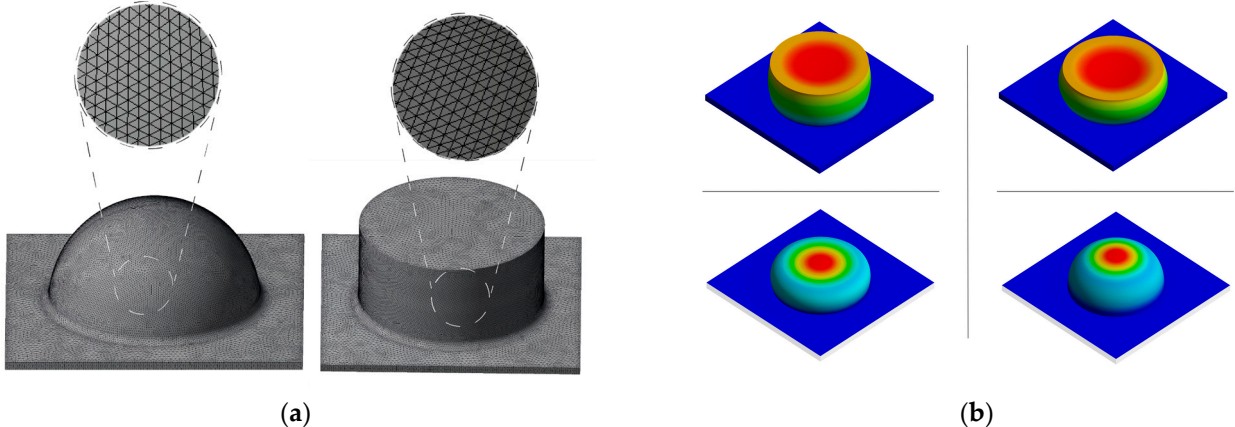

(a)          (b)

**Figure 8.** Sensor cell finite element simulations: (**a**) a mesh study over different shapes of sensor cells; (**b**) stress variation in different load cases for different sensor shapes.

Figure 10 displays the final version of the designed sensor sheet. Initially, the tests were conducted on a 4 × 10 cell sensor sheet, which was later upgraded to the final version of a 6 × 12 cell sensor sheet.

*4.3. Discussion*

4.3.1. Technical Comparison

The findings from the sensor sheet design and integration process can be compared and contrasted with previous studies in the field. The innovative aspect of the SENSO-MATT sensor sheet, which enables pressure measurement from underneath a mattress, is a magnificent achievement.

In Figure 11, for four representative poses of a volunteer, the findings are compared between a SENSOMATT sensor sheet and a commercially available flexible piezoelectric sensor sheet with an array of 27 × 64 components, with each sensor having a normal range of 0 to 100 mmHg for use in mattress applications. Visibly, when the commercial sensor

sheet is placed on top of the mattress, the pressure results are satisfactory, while, when it is placed underneath the mattress, there is no way to detect the body organs, maximum pressure, or even the attitude.

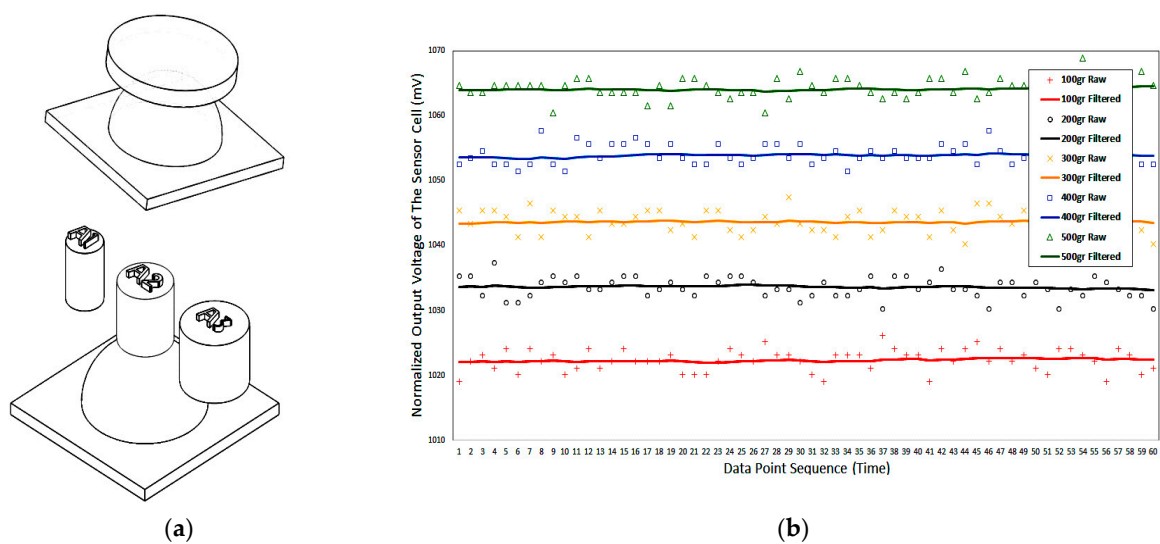

(**a**)  (**b**)

**Figure 9.** Linearity and repeatability of sensor cells while testing with scales. In (**a**) different scales are applied on top of each sensor cellm while the results can be seen in (**b**).

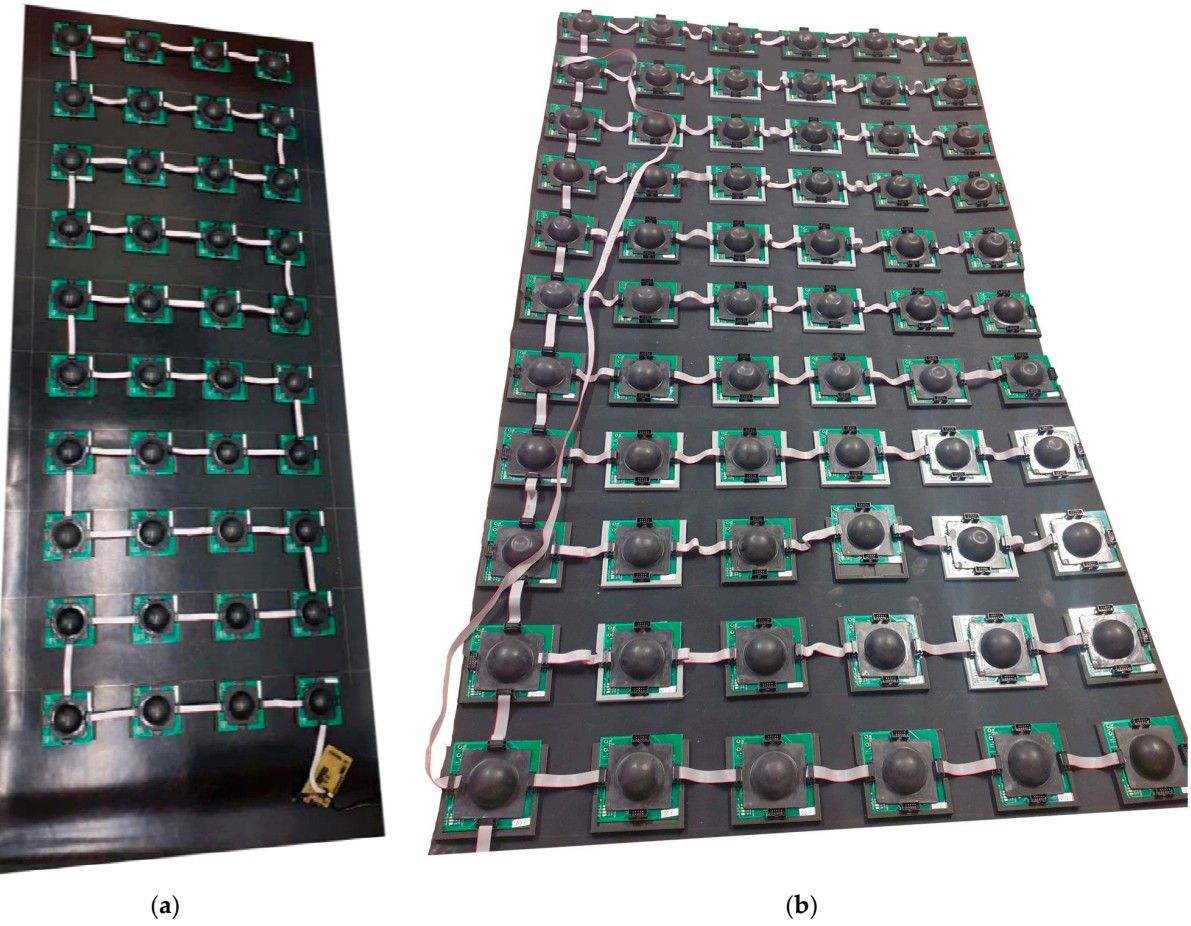

(**a**)  (**b**)

**Figure 10.** (**a**) Completed sensor sheet with 4 × 10 sensor array. (**b**) The same sensor sheet with extended dimensions and 6 × 12 sensor array.

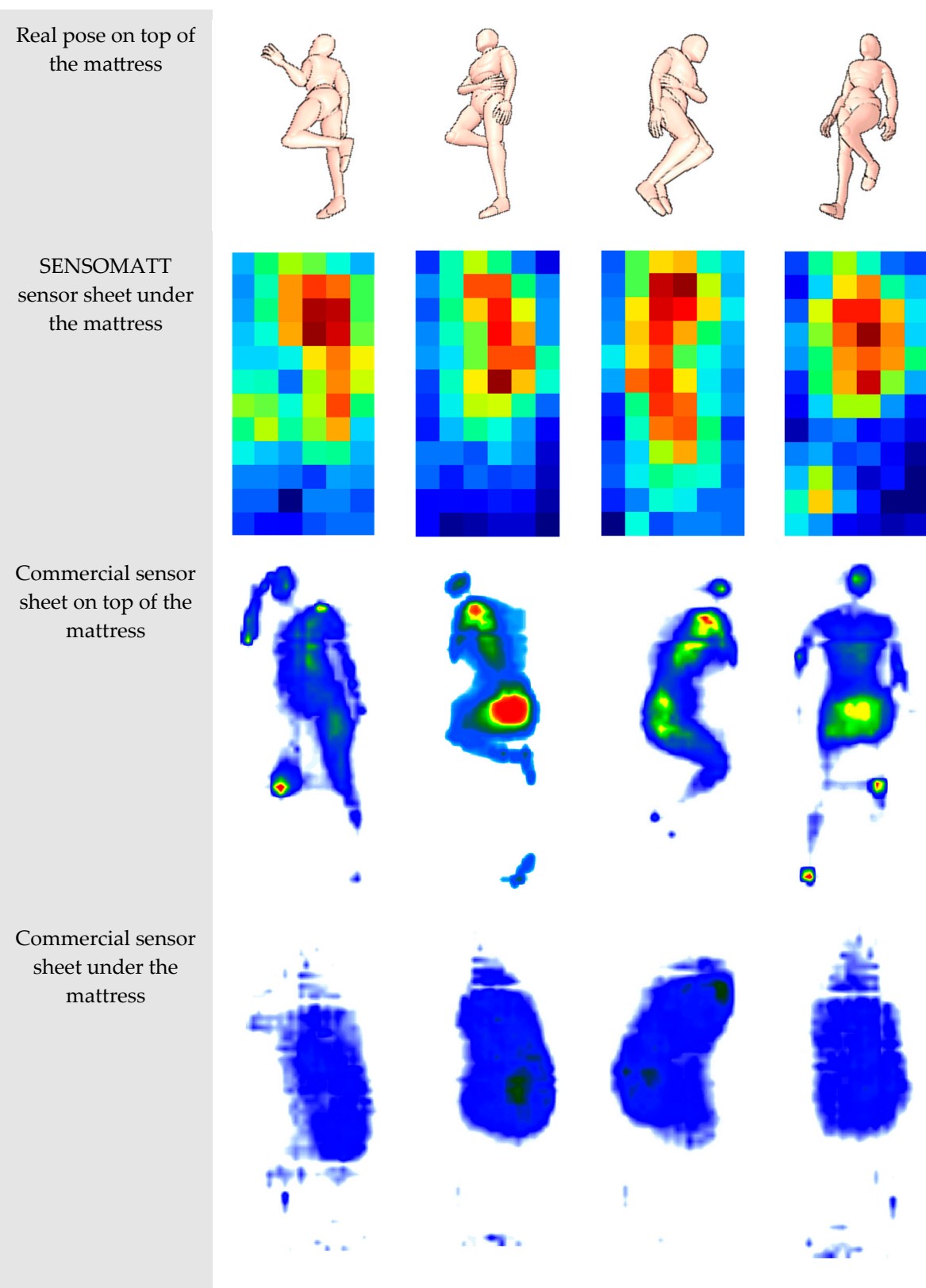

**Figure 11.** The comparison of SENSOMATT sensor sheet results with a commercial elastic piezoelectric sensor sheet when placed underneath the mattress.

The results indicate that pressure mapping using commercial sensor sheets from below the mattress is not viable. This is due to the fact that piezoelectric sensor sheets assess

planar strain, and no significant planar strain is measured beneath a mattress. In contrast, the SENSOMATT sensor sheet can detect meaningful pressure differences beneath the mattress. Due to the omnidirectional nature of its sensors, it is able to measure vertical pressure in addition to horizontal pressure. Fonseca et al. (2023) presented the findings of a comparison between the SENSOMATT sensor sheet and commercial pressure mapping on top of the mattress for 60 volunteers in 28 typical ICU department positions [20].

### 4.3.2. Limitations and Cost Comparison

While the SENSOMATT sensor sheet demonstrated promising results, it is important to acknowledge its limitations and areas for improvement. One limitation could be the cost associated with the production of the sensor sheet, which may hinder its widespread adoption. Future research could focus on optimizing the manufacturing process to reduce costs without compromising the sheet's performance. Despite the SENSOMATT sensor sheet being the most affordable product in the market regarding cost, as depicted in Figure 12, there is still a necessity for further cost reduction. Additionally, the calibration process for different mattresses could be further streamlined to enhance user convenience. Exploring alternative materials or incorporating advanced sensing technologies could also be potential avenues for future development.

| Brand | Country | Dimensions | Price | Works under Mattress |
|---|---|---|---|---|
| BPM | Australia | 180×90 mm | 5375 $ | NO |
| TACTILUS | USA | 180×78 mm | 7200 $ | NO |
| Sxinen | China | 185×85 mm | 6369 $ | NO |
| BodiTrak | Canada | 203×91 mm | 5000 $ | NO |
| XSENSOR | USA | 220×99 mm | 9750 $ | NO |
| SENSOMATT | Portugal | 180×75 mm | 1200 $ | YES |

**Figure 12.** Cost comparison of several commercial pressure mapping sensor sheets on the market (costs may vary depending on the selected store and the date that the data were collected).

### 4.3.3. Clinical Relevance and Impact

The physical presence of sensors on top of a mattress might cause discomfort or skin irritation, especially for users with sensitive skin or those lying on the bed for extended periods. This article speaks about limitations of the sensors: "Owing to their deformability, lightness, portability, and adaptability, flexible skin-like sensors have achieved many functions that were previously unattainable for traditional sensors" [21]. By permitting pressure measurement from beneath a mattress, the sensor sheet provides a non-invasive and practical method for monitoring patients at risk for developing pressure injuries [22]. Early detection and prevention of pressure injuries can improve patient outcomes and minimize healthcare expenditures.

Innovative in design and functionality, the SENSOMATT sensor sheet is poised to change the field of pressure injury prediction and prevention. The non-invasive insertion of the sensor sheet beneath the mattress promotes patient comfort, while the continuous pressure mapping enables early diagnosis of potential pressure injuries.

This device can greatly reduce the incidence of pressure ulcers by enabling healthcare providers to correctly detect high-pressure locations and informing timely intervention. Its versatility with diverse mattress sizes contributes to its therapeutic utility in a variety of healthcare settings, from hospitals to private homes.

In addition, the incorporation of AI technology provides a more refined approach to pressure mapping. The AI system, which was developed using data from various sleeping positions, boosts the sensor sheet's capacity to provide a thorough overview of

pressure distribution. This could significantly improve patient care by providing healthcare personnel with more exact data on which to make their judgments. The integrated AI system could be programmed to suggest individualized interventions based on patient-specific data, promoting a more tailored, patient-centered approach to care [23].

Moreover, the SENSOMATT sensor sheet's cost-effectiveness makes it a realistic alternative for mass adoption, solving the financial constraints that many healthcare facilities confront when deploying new technology. The sensor sheet's data could also be a valuable tool for training, helping to better understand pressure distribution patterns and their relation to pressure injury risk; it could improve their capacity to deliver effective, high-quality care [24].

The impact of the SENSOMATT sensor sheet goes beyond the avoidance of pressure injuries. Continuous and accurate pressure monitoring could be useful in different clinical settings, such as post-operative care, rehabilitation, and long-term care for bedridden patients. In addition, it gives vital data for studies on pressure injuries and other related health conditions, which could contribute to future advancements in this sector.

### 4.3.4. Pressure Injury Prediction Using Pressure Monitoring

In response to the requirement for a non-contact pressure measuring device, the SENSOMATT sensor sheet emerges as a viable solution. However, the measurement process is indirect, yielding critical pressure states determined by a pre-trained AI decision-making core. This AI engine, developed during the SENSOMATT project, will be detailed in a forthcoming journal paper. The decision-making process based on deep data learning involves at least two CNN layers, the first for identifying and extracting key parameters, and the second for deriving these parameters from the SENSOMATT pressure results. In this regard, there are additional possible neural networks designed for fine-tuning and refining the results.

The SENSOMATT sensor sheet comprises 72 sensor data points organized in a $6 \times 12$ matrix format. Raw data are stored in a local memory buffer placed on a central device, not on all sensor cells, or transmitted via Bluetooth to an external storage designed for this purpose. These data serve both for AI training and, in the final product, for result evaluation according to user requirements. Individually, these pressure data points are insufficient for assessing body pressure states. To address this, a pose-based algorithm and a segment-based neural method were concurrently applied and trained using data from 60 volunteers. These data are synchronized with pressure mapping data collected from the SENSOMATT sensor sheet. The neural network, trained using pressure mapping data, discerns the 13 crucial body joints. It identifies the maximum pressure on each joint, accounting for joint proximity and varying effective radii based on joint locations. This information is stored as metadata, shaping the second round of decision making. Each SENSOMATT sensor array result is linked to a pressure mapping outcome, along with a joint–link diagram highlighting the 13 critical joints labeled with their maximum pressure values. A total of 60 volunteers executed 28 distinct poses under three loading conditions, generating 5040 fully labeled datasets for training the second decision-making neural network. The accuracy of this method in measuring the maximum pressure across the 13 joints based on SENSOMATT output data in the test phase surpassed 80%, with further data collection expected to enhance this figure.

In addition to pressure measurement, this approach can be leveraged to assess patient pose and rotation. This assessment accounts for not only current sensor data but also the movement history of each patient. Given anatomical constraints, certain joint rotations are limited; hence, a patient's movement history reveals prone–supine transitions, with any deviations from this pattern documented as patient history.

This technology proves invaluable for predicting pressure injuries. Nevertheless, refining the critical pressure thresholds for each joint necessitates extensive data collection within a clinical environment, alongside expert observations, and patients' medical histories. Our ongoing efforts are directed toward this crucial stage, and the outcomes of the clinical trial for the sensor sheet will be soon available for publication.

## 5. Conclusions and Future Work

This paper introduced the elastic sensor sheet SENSOMATT, a substantial contribution to the prognosis and prevention of pressure injuries. The innovative design of the sheet permits non-invasive pressure measuring beneath a mattress, overcoming the limits of conventional external pressure mapping methods. This design enables adaptability to various mattress sizes with a brief calibration procedure, providing versatility and ease for a wide range of applications. Tests and simulations validated its precision, sturdiness, and dependability in detecting pressure peaks. The SENSOMATT sensor sheet is a cost-effective, continuous monitoring device that enables healthcare providers to identify high-pressure locations and avoid the onset of pressure injuries. Its versatility and user-friendliness make it a helpful tool in a variety of healthcare settings.

- Future work on the SENSOMATT sensor sheet will concentrate on a number of areas for enhancement. Clinical validation is required to establish the usefulness of the sheet in real-world healthcare settings, reducing pressure injuries in varied patient populations. The optimization of production costs can allow broader technological adoption. Utilizing sophisticated technology or alternative materials can improve the performance of the sensor sheet, potentially including novel sensor designs for increased pressure data gathering. Developing wireless communication capabilities and merging data with electronic health records or monitoring systems could provide seamless data delivery. A more intuitive user interface and more powerful data analysis tools can provide greater insight into patterns of pressure distribution. Lastly, it is necessary to analyze the viability of long-term monitoring using the sensor sheet, taking into account power consumption, data storage capacity, and maintenance requirements. These innovations can improve the prevention of pressure injuries and patient care in healthcare settings.

## 6. Patent(s)

- The sensors used in this sensor sheet are patented in Portugal under the national patent organization regulations (INPI) with registration number 117507. The patent is also under PCT international protection with WIPO No. PCT/IB2022/059658. The patent belongs to Sensomatt Lda.

**Author Contributions:** Conceptualization, M.M.A., M.G.F.D. and D.F.S.; methodology, M.M.A. and M.G.F.D.; validation, M.M.A., D.F.S., M.F. and M.G.F.D.; formal analysis, M.F. and R.D.; investigation, D.F.S., M.G.F.D., P.A., F.H., M.F., F.R., J.C.M. and L.F.; resources, P.A. and R.D.; data curation, D.F.S., M.G.F.D., F.R., J.C.M. and L.F.; writing—original draft preparation, M.M.A., M.G.F.D., P.A. and F.H.; writing—review and editing, M.G.F.D., P.A. and R.D.; supervision, M.M.A., P.A. and M.F.; project administration, M.M.A. and D.F.S.; funding acquisition, M.M.A. All authors have read and agreed to the published version of the manuscript.

**Funding:** This work was carried out under the SensoMatt project, grant agreement no. CENTRO-01-0247-FEDER-070107, co-financed by European Funds (FEDER) by CENTRO2020.

**Data Availability Statement:** Data collected by using the sensor sheet introduced in this article could be found here: https://github.com/rdionisio1403/PoPu/. Proper consent is taken from volunteers and the published data does not contain any kind of sensitive information, or information which may lead to the volunteer's identification.

**Conflicts of Interest:** The authors declare no conflict of interest.

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
