# Peer review of "A Novel Elastic Sensor Sheet for Pressure Injury Monitoring: Design, Integration, and Performance Analysis"

_electronics, doi:10.3390/electronics12173655_

Round 1

Reviewer 1 Report

In this paper (electronics-2483225), the authors exhibited the design and integration of the SENSOMATT elastic sensor sheet for pressure ulcer prediction. The application object is interesting, and the strategy are basically complete, however, there are some problems in the writing, experiment, mechanism analysis, and necessary data results. Some revisions need to be addressed before possible publication.
1. The title does not reflect innovation well, it is recommended to provide detection target (ulcer prediction).
2. Introduction: What are the current problems faced by flexible pressure sensors/arrays? What is the substantive innovation of this work? Suggest analyzing and discussing the current research status of flexible pressure sensors (may refer to Electronics 2023, 12(8), 1782; J. Mater. Chem. C, 2023, 11, 5585–5600), and comparing them with SENSOMATT sensor to highlight its advantages.
3. It is recommended to merge the literature review in the second part into the introduction. There are only 7 references [3-7], which cannot fully reflect the current research status.
4. “SENSOMATT sensor”, the parameter information of the sensor, especially the pressure sensing performance indicators, needs to be provided.
5. Please briefly describe the working principle and advantages of the SENSOMATT sensor.
6. Results (section 7): Application test results data (for example, simulation applications) should be provided.
7. Discussion (section 8): Specific data results need to be provided for comparison and evaluation
8. A space is required between numbers and units.
9. The number of references is too small to fully reflect the current research status.
10. Check English writing.

 Minor editing of English language required

Reviewer 2 Report

The idea is quite interesting. However, the content is not systematic. And also, I can not find any significant presented data or graph to show the performance of their sensors. 

- Please expand and elaborate the content of abstract to include the main background or motivation, as well as compact the primary experimental method and main finding from this study.

- In the introduction part, it is important to detail the information of current existing technology to prevent the pressure ulcers. The second paragraph of introduction is quite general and not informative enough about the established literature

- can the authors make a table of comparison for their design compared to previous works in term of the cost, invasiveness, and the operation system?

- Please improve and modify the materials and methods section. Make it into several sub section and discuss in more detail about the technical experimental information and the parameters, etc. This section could be emerged with the section 4-6.

- which data and graph discussing about the result in section 6.2 and 6.3 (it is typo in the manuscript; it should be 7.2 and 7.3)?. The result and discussion are really poor and limited. 

- There is not significant data or graph in the main figures

Reviewer 3 Report

The topic of the subject to which the present study is addressed is challenging in the medical field, as a result of the prevalence of pressure ulcers and the implications and complications following their installation.

The authors present an attractive technical solution, using a non-invasive network of sensors placed under the mattress that allows the patient, their relatives or the medical staff to intervene effectively and appropriately in situations with a high risk of pressure ulcers.

One of the most significant drawbacks of this article is specifically the performance evaluation section, as follows:

• Although it was mentioned that a series of tests were carried out to evaluate durability, accuracy and sustainability, nevertheless a quantitative expression of them was not presented in the article, but only a qualitative statement.

• Extensive testing and clinical trials are also mentioned, whose descriptions and results are not indicated. What biological parameters are measured, and how can they help reduce pressure ulcers?

• It is stated, "The network's ability to transmit data accurately and efficiently was evaluated through comprehensive testing". What are these tests and what are their results

Although several similar products are mentioned in the introduction, the discussion section does not say what is innovative in this product, nor are the comparative advantages and disadvantages presented by the proposed sensor sheet described.

Consequently, I recommend a major revision in which to take into account the requirements mentioned above.

Reviewer 4 Report

The work is interesting, so for its publication, the authors must clarify in detail the following issues:

- Does the sensor have a memory of each body area?

- How does the sensor know the total pressure accumulated in each body area?

- If the patient turns or turns upside down, how does the sensor know that the accumulated pressure corresponds to another body area? Is the sensor sensitive to rotation?

- If the sensor has memory, from what threshold would the risk of ulcers increase?

- Is there an algorithm that indicates to the sensor the risk of ulcer appearance or that the accumulated pressure in any body area is high and is associated with the risk of ulcers?

- The authors have applied these sensors in some population, to really check their usefulness and if they produce any kind of inconveniences (discomfort, allergies, etc.).

Round 2

Reviewer 1 Report

The revised manuscript is satisfactory and recommended for publication.

Author Response

Thank you very much for your kind attention and invaluable comments to improve our article.

Reviewer 2 Report

Thanks to the Authors for their efforts in improving the manuscript with a lot of additional data and description. The Authors have addressed all my suggestion and concern. I support for the publication

Author Response

(The authors gave the same response as above.)

Reviewer 3 Report

The majority of the review requirements have been outlined by the authors. The article is likely to be approved with only a few minor adjustments, which include the following:

- Capitalizing the first letter in the subtitle "Practical Results" in section 4.1 on line 344.

- Reloading Figure 7 as it is currently not in a visible format.

Author Response

Thank you very much for your kind attention and invaluable comments to improve our article.
We considered all your suggested the minor changes and once again checked everything to avoid typo and writing defects.

Reviewer 4 Report

The authors have left many questions unresolved, regarding the memory and accumulated memory of the sensors according to the different body areas, as well as their sensitivity if patients rotate in bed. Nor do they clarify the thresholds, from which the risk of an ulcer in a body area would appear. In my opinion, more research is needed on the issues discussed above, to be able to say that this technology is really useful in the prevention of pressure ulcers.

Author Response

Dear Sir,

Thank you very much for your kind attention and invaluable comments.

I apologize for any inconvenience caused. In our revised paper, we inadvertently omitted some of the answers regarding the sensor memory and data storage. These answers were, however, included in the answers document. We have now incorporated them into the revised paper, highlighted in green for your convenience.

Regarding your inquiries, we have introduced a new subsection, 4.3.4, titled "Pressure Injury Prediction Using Pressure Monitoring," spanning nearly a page. It's important to note that the pressure measurement based on our sensors' signals is not direct in nature. Therefore, the displayed pressure values do not directly indicate the pressure on body organs. Instead, our approach involves a deep data learning platform that employs a pre-trained algorithm utilizing the 72-array matrix to assess potentially risky situations. This method parallels the detection of patient poses and rotations. The sensors themselves do not measure values directly.

Elaborative details of the CNN algorithm employed, the data refinement process, accuracy enhancement techniques, and dataset loading procedures are expounded in a separate AI-focused paper solely dedicated to this subject matter. In terms of the technology's usefulness, it's important to acknowledge that any medical device must undergo clinical trial stages to ascertain its effectiveness. Currently, our technology is undergoing its initial clinical trial, the results of which are slated for publication in a related medical journal within the upcoming 3-4 months.

The current paper primarily serves to illustrate the methodology behind developing an electronic hardware for data collection, facilitating a secondary decision-making layer centered around a deep learning core. It showcases our achievements in creating hardware capable of reliably collecting data from beneath a mattress, surpassing the capabilities of conventional planar piezoelectric sensor sheets. However, for in-depth discussions on decision-making accuracy and body pressure measurement, a separate AI-centric paper delineates the progressive refinement of our algorithms.

Regarding the technology's application in hospitals and medical centers, particularly in the prevention of pressure injuries, we wish to highlight that the ongoing clinical trial will contribute to gathering relevant data and subsequent publication of related research.

Thank you once again for your feedback and guidance.

Best Regards, 

Mohammad Mohammad Amini
